# A Comparative Study on the Meat Quality, Taste and Aroma Related Compounds between Korean Hanwoo and Chikso Cattle

**DOI:** 10.3390/foods12040805

**Published:** 2023-02-13

**Authors:** Van-Ba Hoa, Dong-Heon Song, Kuk-Hwan Seol, Sun-Moon Kang, Hyun-Wook Kim, In-Seon Bae, Eun-Sung Kim, Yeon-Soo Park, Soo-Hyun Cho

**Affiliations:** 1Animal Products Utilization Division, National Institute of Animal Science, RDA, Wanju 55365, Republic of Korea; 2Jeonbuk Livestock Research Center, Jinan-Gun 55460, Republic of Korea; 3Gangwon-do Livestock Research Institute, Hoengseong-Gun 25266, Republic of Korea

**Keywords:** breed, Hanwoo, Chikso, meat quality, aroma, taste

## Abstract

The aim of this study was to compare the meat quality and taste-and-aroma-related components of beef between breeds. For this purpose, Hanwoo and Chikso steers (*n* = 7 per breed) raised under identical conditions until 30 months old were used. After 24 h of slaughter, *longissimus lumborum* (LL) and *semimembranosus* (SM) muscles were collected and analyzed for technological quality, free amino acids, metabolites, and volatile compounds. The Chikso meat showed lower values for shear force and color traits (lightness, redness, and yellowness) compared to Hanwoo (*p* < 0.05). The Chikso presented a higher amount of sweetness-related free amino acids (alanine, proline, and threonine) in the LL muscle, while Hanwoo had a higher amount of methionine and glutamine associated with umami taste (*p* < 0.05). A total of 36 metabolites were identified and quantified in the meat samples; out of them, 7 compounds were affected by breed (*p* < 0.05). Regarding aroma compounds, a significantly higher amount of fat-derived aldehydes associated with fatty and sweet notes was found in Hanwoo, whereas a higher amount of pyrazines associated with roasty notes was found in Chikso (*p* < 0.05). Thus, under identical feeding conditions, breed showed a significant effect on the quality and taste-and-aroma-related components that may influence the eating quality of beef between the two breeds studied.

## 1. Introduction

The Chikso breed, together with three other Korean native cattle breeds, has been registered with the Domestic Animal Diversity Information System of the Food and Agriculture Organization. The Chikso breed is characterized by its unique brindle coat color [1], which is completely different from the other registered cattle breeds (e.g., brown Hanwoo, Jeju black cattle, and Korean black cattle). Historically, the Chikso breed was used mainly as draft and pack animals. Compared to the other popular commercial beef cattle breeds (e.g., Hanwoo) raised in Korea, the Chikso breed is generally maintained at a smaller population size, around 4000 heads in 2016 [2]. In recent years, due to the increasing demand for safe meat products derived from native cattle breeds in South Korea, the Chikso breed has been recognized as a valuable breed and has received more attention from beef producers [3]. In contrast to the Hanwoo breed, which is commonly raised following the standardized production process and fed grain feed in feedlots of commercial farms, the Chikso breed is currently raised by small-scale producers in several localities (e.g., Gyeongbuk and Kangwon provinces) in the country [4]. Until now, almost all studies on the Chikso breed have only focused on the genetic diversity aspects [3,4].

Meat quality can be defined in different ways, from product yield to a set of properties (e.g., color, texture, water holding capacity, eating attributes, etc.) that together identify what we appreciate about meat when buying or eating it and using it as raw material for processing into meat products [5]. It is well known that there are many factors influencing the quality of meat [6]. Among others, animal characteristics (e.g., genetics or breed and feeding diets) significantly affect the physiochemical composition of muscle tissues, which subsequently affects the meat quality [7].

More to the point, flavor, consisting of odors and tastes, is among the leading factors determining the eating quality of meat and its products by consumers [8,9]. Tastes (sweetness, saltiness, bitterness, sourness, and umami) of cooked meat are contributed by non-volatile constituents or taste-active compounds (free amino acids and metabolites) of fresh meat [10,11,12]. Otherwise, these constituents, together with others (lipids), are the major contributors to the formation of aroma characteristics via the Mallard reaction and/or thermal oxidations during the cooking/heating of meat [13]. Researchers have discovered that both pre- and post-harvest management can influence the flavor precursors of fresh meat and, as a result, the flavor characteristics of cooked meat [14].

To the best of our knowledge, limited scientific information regarding the quality characteristics of Chikso beef is available. Also, there is a lack of comparative studies on meat quality between this breed and other commercial cattle breeds under identical rearing conditions. Therefore, the aim of this study was to determine the meat quality, taste, and aroma-related components of Chikso beef and compare them with those of Hanwoo beef under identical raising conditions.

## 2. Materials and Methods

### 2.1. Sampling

A total of 14 steers (7 for Hanwoo and 7 for Chikso) were used in the present investigation. All the animals were raised under identical commercial conditions and fattened in feedlots with a grain-based diet. During the growing phase, they were fed twice/day with formula feed (3.0–7.5 kg) and rice straw (3.0–4.0 kg). At the fattening phase (from 18 months of age), they were fed *ad libitum* with a high rate (90%) of concentrate feed (11–12% crude protein) and only 10% rice straw up to 30 months of age. The average body weight at slaughter was around 680 and 740 kg for the Chikso and Hanwoo steers, respectively. The animals were slaughtered at an abattoir of the National Institute of Animal Science (NIAS), Wanju-gun, Korea. All procedures were approved by the Institutional Animal Care and Use Committee (IACUC) of NIAS (Approval No. NIAS 20001992). The next day (24 h post-mortem), two representative muscles including: *Longissimus lumborum* (LL) and *semimembranosus* (SM) (*n* = 7 per breed and muscle type) were removed from the left side of all the carcasses and immediately used for analyses. After trimming all visual fat and connective tissues, each muscle was cut into sub-samples depending on the analysis. The samples for analysis of quality attributes (color, pH, water holding capacity, cooking loss, and shear force) were immediately used after cutting, while the rest of the meat samples were stored at −20 °C for free amino acids, metabolites, and aroma-flavor compound analysis.

### 2.2. Meat Color and pH Measurement

The color and pH were measured on the same samples (in the form of 2.5-cm-thick steaks). For the color, it was measured on 5 different areas on the transverse section of each sample using a Minolta Chroma Meter CR-400 with a D65 illuminant and 2° observer (Minolta Camera Co., Osaka, Japan). Prior to the measurement, all the samples were kept at 4 °C for blooming for 30 min. The results were expressed as CIE L* (lightness), CIE a* (redness), and CIE b* (yellowness). Immediately after the color measurement, the pH was measured in triplicate by inserting the solid-state probe of the pH meter (pH*K 21 m, NWK-Technology GmbH, Kaufering, Germany) deeply into the meat samples. Before use, the pH device was calibrated with pH 4.0 and 7.0 standard solutions (NWK Technology, City, Country, Germany).

### 2.3. Cooking Loss and Warner-Bratzler Shear Force (WBSF)

The cooking loss and WBSF were measured on the same piece of each sample according to the method described by Hoa et al. [15]. Briefly, a steak (weighing 200 g and having a 2.5 cm thickness) was taken from each muscle sample, trimmed of its outer fat, placed in a polyethylene bag, and cooked in a 73 °C preheated water bath until the core temperature reached 72 °C. After cooking, the cooked samples were cooled in ice water for 30 min, removed from the bags, and blotted dry with paper towels. The sample weight before and after cooking was recorded to determine the cooking loss (pre-cooking weight minus post-cooking weight divided by the pre-cooking weight and multiplied by 100). The WBSF of cooked meat samples was measured using an Instron Universal Testing Machine (Model 4465, Instron Corp., High Wycombe, UK) at a crosshead speed of 200 mm/min and a 50 kg load cell. For this, 5 strips per cooked sample were made parallel to the muscle fiber direction using a 0.5-inch metal corer. WBSF values (kg-force) were obtained by completely cutting the strips with the device.

### 2.4. Water Holding Capacity (WHC)

The WHC of each meat sample was measured in duplicate using the procedure as described by Hoa et al. [15]. Briefly, after grinding using a mini grinder (Hanil Co., Chungcheongnam-do, Korea), an aliquot (0.5 g) of meat was taken and placed in a 2 mL ultra-centrifugal filter unit, which was then inserted into an ultra-centrifugal filter device (Millipore Corp., Bedford, MA, USA). After heating for 20 min at 80 °C in a water bath, the samples were cooled at 4 °C for 10 min and then centrifuged at 2000× *g* for 10 min. The initial weight of the ultracentrifugal filter unit before cooking and its weight with the cooked sample were recorded to determine the water loss. Also, the total moisture and fat contents in each fresh meat sample, determined by using a Food Scan Lab 78810 (Foss Tecator Co., Ltd., Hillerod, Denmark), were used to determine its WHC. Finally, the WHC was calculated using the equation developed by Laakkonen et al. [16] as follows:(1)$$\mathrm{WHC}=\frac{\mathrm{Total}\;\mathrm{moisture}\;\mathrm{content}\;\left(\%\right)-\mathrm{Moisture}\;\mathrm{loss}\;{\left(\%\right)}^{1)}}{\mathrm{Total}\;\mathrm{moisture}\;\mathrm{content}\;}\times 100$$
(2)$${\;}^{1)\;}\mathrm{Moisture}\;\mathrm{loss}\;\left(\%\right)=\frac{W1-W2}{\;S\times {\mathrm{FF}}^{2)}\;}\times 100$$
(3)$${\;}^{2)}\;\mathrm{FF}=1-\frac{\mathrm{Total}\;\mathrm{fat}\;\mathrm{content}}{10}$$

W1: Weight of sample and centrifugal filter unit before heating. W2: Weight of sample and centrifugal filter unit after heating and centrifuging. S: Sample weight. ^2)^ FF: Fat factor; 1: Constant.

### 2.5. Tastes-Related Compounds (Free Amino Acids, FAAs and Metabolites)

The content of FAAs in the meat samples was analyzed using the protocol described by Cho et al. [17] with minor modifications. Briefly, 2.5 g of each sample was weighed, placed in a conical tube, and homogenized with 5 mL distilled water at 1200× *g* for 1 min. The homogenate was filtered with Whatman filter paper (Whatman Inc., Clifton, NJ, USA), and an aliquot of 100 µL filtrate was taken and mixed with 900 µL methanol containing 0.1% formic acid. Next, the samples were centrifuged at 13,000× *g* at 4 °C for 10 min, and the supernatant was again filtered through a 0.45-μm filter membrane (Millipore Ltd., Cork, Ireland). After derivatizing with AccQ-Tag^TM^ (Waters Co., Milford, MA, USA), according to the manufacturer’s instrument, the samples were used for FAA analysis. The FAA composition was separated on an amino acid column (2 × 50 mm, 3 μm, Imtaka, Uphur St., Suite A, Portland) connected to a Waters Acquity UPLC (model: Xevo TQ-S, Waters Co., Milford, MA, USA). The solvents used were: A [acetonitrile: 100 mM ammonium formate; 20:80 *v*/*v*] and B [acetonitrile: trifluoroacetic acid: 25 mM ammonium formate; formic acid: 9:75:16:03 *v*/*v*/*v*]. The UPLC conditions were set as follows: Separation temperature at 37 °C, flow rate of 0.4 mL/min, and solvent ingredient: initial 100% B, linear change to 83% B for 6.5 min, linear change to 100% A for 3.5 min, and then linear change to 100% B for 2 min, and maintained for an additional 5 min. The results were expressed as milligrams per 100 g of meat (mg/100 g meat). Each sample was determined in duplicate.

For the analysis of metabolites, the meat samples (2.5 cm thick in steak form) were cooked on a frying pan at around 180 °C for approximately 4 min. During the frying, the meat was turned at 1-min intervals. The extraction and separation of metabolites were carried out following the protocol as described in our previous study [17]. Briefly, triplicate aliquots (20 mg) of meat were weighed and extracted with an acetonitrile/water (1:1, *v*/*v*) mixture on ice for 10 min. After centrifuging at 3000× *g* for 10 min at 4 °C, the supernatant was collected and freeze-dried. The metabolites were analyzed using a 600 MHz Agilent NMR spectrometer (Agilent Technologies, Palo Alto, CA, USA) equipped with a 600 MHz 4-mm gHX NanoProbe (Agilent Technologies, Santa Clara, CA, USA) at a ^1^H frequency of 599.93 MHz. The analytic conditions used were the same as those described by Cho et al. [16], and the acquired ^1^H- spectra were identified using the Chenomx 600 MHz library database and Chenomx NMR Suite 7.1 professional. The concentration (mM/kg) of each metabolite was determined using the known concentration of an international standard (3-trimethylsilyl-2,2,3,3-tetradeuteropropionic acid-d4 (TSP-d4, Sigma-Aldrich, St. Louis, MI, USA). The results were expressed as mM/kg of meat sample.

### 2.6. Volatile Flavor Compounds

The aroma volatile compounds in the meat samples were extracted using a solid-phase micro-extraction (SPME) method as described in our previous work [18]. To strengthen the chemical reactions (e.g., Mallard reaction) for the formation of volatile flavor compounds, the fresh meat samples were ground into small particles before cooking. The ground meat samples were then cooked on a frying pan at around 180 °C for 2 min (the meat was stirred throughout the frying process). Immediately after cooking, the samples (1.0 g each) were taken and placed into a 20-mL headspace vial and tightly capped with a magnetic screw cap. For quantification, 1.0 μL of an internal standard (2-methyl-3-heptanone at 816 mg/mL in methanol) was also added. The extraction of aroma volatiles was done using a 75 µm SPME assembly of CAR/PDMS fiber (black, autosampler type, Supelco, Bellefonte, PA, USA) connected to a SPME auto-sampler (model: PAL RSI 85) of gas chromatography (model: 8890 GC system) and mass spectrophotometry (5977B MS, Agilent Technologies) at 60 °C for 50 min. The extracted volatiles were desorbed at the injection port at 250 °C for 5 min and then separated on a DB-5MS column (30 m × 0.25 mm i.d. × 0.25 μm film thickness; Agilent J & W Scientific, Folcom, CA, USA). The GC/MS conditions set were the same as those used in the above-cited reference [18]. The peaks were identified by comparing their mass spectra with those in the Wiley registry library (Agilent Technologies) and/or by comparing their retention times with those of external standards. The quantification of each compound was carried out by comparing its peak area percentage with that of the internal standard (1.0 μL of 2-methyl-3-heptanone, 816 mg/mL in methanol). The results were expressed as μg/g of meat sample.

### 2.7. Statistical Analysis

Statistical analysis was conducted using SAS Enterprise software (version 7.1; SAS Institute, Inc., Cary, NY, USA). For each muscle type, data was separately analyzed using the General Linear Model procedure, where the cattle breed was considered a fixed effect and the quality attributes, free amino acids, metabolites, and aroma compounds were considered dependent variables. The mean difference was compared using Duncan’s Multiple Range Test, and significant differences were set at a 5% level. Data was presented as means ± standard deviation.

## 3. Results and Discussion

### 3.1. Technological Quality

The mean values for technological quality (pH, water holding capacity, cooking loss, and shear force) of the beef samples from two breeds are shown in Table 1. The ultimate pH of meat is usually determined by the extent of pH decline within 24 h of slaughter. As we know, postmortem glycolysis (the process that converts glycogen into energy under anaerobic conditions) is the main process resulting in increased H^+^ ion accumulation and pH decline in meat. The rate and extent of post-mortem glycolysis vary depending on animal species, genetics, feeding systems, and stress caused during fasting and transport [19]. Researchers have also reported that any abnormal rate of postmortem glycolysis may result in inferior meat quality [20,21]. Our results showed that no differences in pH values occurred between the two cattle breeds (*p* > 0.05). This means that the rate of glycolysis or pH decline was similar in both breeds studied. Warren et al. [22] reported that under identical feeding conditions, breed does not influence the ultimate pH of beef *longissimus lumborum* muscles. Contrastingly, Xie et al. [23] showed a wide variation in pH values of beef *longissimus dorsi* muscles among Limousin, Simmental, Luxi, Qinchuan, and Jinnan.

Water-holding capacity is the ability of fresh meat to retain moisture, which is an important technological quality trait. The results showed that the breed only affected the WHC of LL muscle, with a higher value in the Chikso compared to the Hanwoo (*p* < 0.05). It is reported that ultimate pH is the major factor affecting the WHC of meat; low ultimate pH (near the isoelectric point of proteins) causes protein denaturation, which in turn lowers the WHC [24]. Since the pH showed no difference between the two breeds, there might be other factors (e.g., quality and quantity of protein or types of muscle fiber) affecting the WHC of beef samples rather than the ultimate pH.

Tenderness is an important factor determining the eating quality of meat [6]. For decades, the WBSF has been considered a highly reliable technique for evaluating beef tenderness [25,26]. Our results showed that in both muscles, the Hanwoo had a significantly (*p* < 0.05) lower WBSF value compared to the Chikso. According to the classification of beef tenderness based on the WBFS values by Belew et al. [26], the LL and SM muscles from Hanwoo can be considered “tender” (3.2 < WBSF < 3.9 kg) and “intermediate” (3.9 < WBSF < 4.6 kg) cuts, respectively. Whereas, both the muscles from Chikso can be considered “tough” (WBSF > 4.6 kg) cuts. The lower shear force values in the Hanwoo beef may be attributed to its higher intramuscular fat (IMF) content compared to the Chikso (our analysis showed that the IMF was approximately 3 times higher in the Hanwoo compared to the Chikso), because the level of IMF is negatively correlated to the WBSF values [27,28]. Otherwise, Chikso is associated with increased physical activity, which may result in tougher meat.

### 3.2. Color Traits

Color is known as an indicator of the freshness and wholesomeness of meat, and it largely affects the purchasing decisions of the meat by consumers [21,29]. The values of color traits of LL and SM muscles from Chikso and Hanwoo are presented in Table 2. It was observed that the cattle type affected all the color traits, in which the Chikso meat exhibited lower L* (lightness), a* (redness), and b* (yellowness) values compared to the Hanwoo meat (*p* < 0.05). This signifies that the Chikso meat seemed to be darker compared to the Hanwoo meat. Hughes et al. [30,31] conducted two studies to investigate the light scattering of beef longissimus muscles among color groups; these authors found that the light, medium, and dark groups had the L* (lightness) values of 35.3–37.3, 34.5–35.0, and 28.2–29.7, respectively. According to the beef classification based on color by these authors, Hanwoo and Chikso meat belonged to the light and medium color groups, respectively. Swatland [32] reported that dark beef usually has a swollen muscle fiber structure with less light scattering compared to light muscles. On the other hand, the color of beef is strongly affected by genetics, the feeding system, and its chemical composition (e.g., level of fat, moisture, and pigment myoglobin) [33,34,35]. Beef meat with a higher fat content is associated with a lighter and redder color [36]. According to our analysis, the IMF content in the Hanwoo was approximately 3 times greater than that in the Chikso. Therefore, the greater fat content of the Hanwoo may be responsible for its lighter color.

Furthermore, the red color of meat is mainly determined by the pigment myoglobin and its biochemical states [29,34] as well as the ultimate pH [30]. In the present study, all the samples were collected at the same time (24 h postmortem), at the same temperature (4 °C), bloomed for the same period of time (30 min), handled under identical conditions, and had similar pH values (Table 1). The paradoxical color results, therefore, may be due to the differences in muscle fiber structures, fat content, and pigment proteins between the two cattle breeds. Similar to the current finding, Aviles et al. [33] reported that breed has an effect on beef color in that local cattle breeds usually present darker meat with a higher haematin content (a dark blue pigment) compared to meat production-specialized breeds.

### 3.3. Free Amino Acids (FAA) and Metabolites

Taste is an important sensory trait of muscle foods. Meat tastes include saltiness, sweetness, sourness, and umami, and FAAs are known as meat’s taste-active components [37,38]. The concentration of FAAs in beef from Chikso and Hanwoo is presented in Table 3. It was observed that alanine, glutamine, leucine, and glutamate were the most predominant FAAs in both the muscles from Chikso and Hanwoo. Supporting the present results, Jayasena et al. [39] and Cho et al. [40] reported a similar trend for these FAAs in the *longissimus* muscles of Hanwoo steers. Based on their similar taste qualities, Kato et al. [40], Dashdorj et al. [41], and Frank et al. [12] categorized the FAAs into different classes such as sweetness (threonine, glycine, alanine, serine, and proline) and umami taste (alanine, glutamine, glutamic acid, aspartic acid, lysine, methionine, and serine). Results show that in the LL muscle, alanine, proline, and threonine, and in the SM muscle, methionine and glutamine, were affected by the breed (*p* < 0.05). The concentrations of alanine, proline, and threonine (associated with sweet taste) were higher in Chikso’s LL muscle, while methionine and glutamine (associated with umami taste) were higher in Hanwoo’s SM muscle (*p* 0.05). Similar to the present results, Koutsidis et al. [11] and Frank et al. [12] found an effect of breed on FAAs in the LL beef muscle. In a study conducted by Dashdorj et al. [41], nearly half of FAAs in beef *longissimus dorsi* muscles were affected by breed (Hanwoo and Angus) when using different feeding diets. In a recent study conducted by Cho et al. [17], almost all FAAs detected in the same beef muscles were affected by gender, with meat from cows having a higher FAA content compared to steer meat.

Metabolites are the final or intermediate products of the metabolic process in muscle tissues after slaughter, and many of them have been reported to contribute to the flavors of or act as precursors for the development of aromas in cooked meat [42,43]. The representative image of the ^1^H NMR spectrum showing the presence of polar metabolites in the LL muscles of both cattle breeds is shown in Figure 1. In addition, as shown in Table 4, a total of 36 metabolites were identified and quantified in both muscles of two breeds. Out of them, lactate was the most predominant metabolite (261–236 mmol/kg), followed by creatine, carnosine, and glucose in both the muscles of the two breeds. Lactate and glucose have been reported to be formed from the glycolytic pathway in post-mortem muscles [44]. Adenosine 5′-monophosphate (AMP), inosine, and hypoxanthine are the nucleotide-derived metabolites formed from the breakdown of adenosine triphosphate (ATP) by endogenous enzymes, while carnitine, carnosine, creatine, and creatinine are formed from proteolysis [45]. The remaining compounds were amino acids, which are also formed from the proteolytic process by endogenous enzymes in meat [45]. Our statistical analysis shows that the breed affected the concentrations of 4 compounds (hypoxanthine, inosine, phenylalanine, and O-acetylcarnitine) in LL muscles, with a higher amount for the Hanwoo (*p* < 0.05). In the SM muscle, anserine, isoleucine, and trimethylamine were affected by the breeds (*p* < 0.05). Almost all of the metabolites found in our samples have also been reported in beef LL muscles from various breeds (Hanwoo and Santa Gertrudis-Brahman cross steers) [15,17,45] and pork [46]. Research conducted to examine the effect of sex metabolites in Hanwoo beef has shown that cow meat has higher acetate, creatine, creatinine, glycine, tyrosine, etc. than steer meat [17]. Hoa et al. [15] showed that the castration method also affects the metabolites in Hanwoo LL and SM beef muscles. Thus, this study examined for the first time the effect of breed on the FAAs and metabolites, with the results indicating that the variations in their concentrations may lead to a difference in the quality of cooked meat between the two breeds.

### 3.4. Volatile Flavor Compounds

As a part of flavor, aroma (sensed by the nose) is among the most important factors determining the eating quality of cooked meat [9]. Survey studies on customer satisfaction for beef have revealed that flavor and tenderness contribute equally to overall like ratings [47,48]. The aroma and flavor are contributed by a variety of volatile compounds that are formed from various chemical reaction processes such as the thermal oxidation of fat, the Mallard reaction between amino acids and reducing sugars, and the interaction of intermediates between these two pathways during the cooking/heating of meat [13]. Studies have stated that the quality and quantity of volatile flavor compounds are largely affected by the contents of precursors (e.g., amino acids, sugars, fat, etc.) in the raw meat [9,11]. The concentration (μg/g) of volatile compounds in the cooked Chikso and Hanwoo meat samples is shown in Table 5. A total of 47 compounds—21 aldehydes, 7 alcohols, 4 ketones, 6 sulfur- and nitrogen-containing compounds, and 9 hydrocarbons—were detected and identified in the cooked LL muscles from both breeds. It is well recognized that almost all aldehydes (except the Strecker aldehydes) are formed in cooked meat from the thermal oxidation of fatty acids during cooking [13,49,50]. With their low odor detection thresholds, aldehydes appear to be important contributors to cooked meat aromas [9,13]. Out of the aldehydes, 8 compounds, including pentanal, hexanal, E,2-hexenal, heptanal, benzaldehyde, E,E,2,4-nonadienal, nonanal, and decanal, were affected by the breed, with a significantly (*p* < 0.05) higher amount in the Hanwoo compared to the Chikso. These aldehydes have been characterized as exerting pleasant aromas such as fatty, sweet, and fruity odor notes in cooked pork and beef [8,12]. The higher aldehyde content in Hanwoo beef could be the result of its higher IMF content, as mentioned above. Because the thermal oxidation of fat is the major pathway for the formation of these aldehydes during cooking [49,50], similar to the current finding, Hoa et al. [15,27] reported that beef with a higher IMF content has a higher amount of volatile flavor compounds. According to research on the effects of breed on flavor compounds of cooked beef, Wagyu beef with a higher IMF content has more fat-derived flavor compounds associated with fatty and sweet odors than the lower-fat Angus beef breed [12]. Beside the fat-derived aldehydes, some of the identified compounds (e.g., 2-methyl pentanal, 2-methyl propanal, 3-methyl butanal, and 2-methyl butanal) are known to be formed from the Strecker degradation of amino acids (e.g., isoleucine and leucine) during the cooking of meat [13,49]. Our result shows that all of these Strecker aldehydes were not affected by the breed (*p* > 0.05). This is probably due to the similar amounts of their corresponding precursors, such as FAAs (Table 3), and metabolites (Table 4), between the two breeds.

Alcohols, which have a low odor detection threshold, contribute significantly to the development of cooked meat flavor [8,13]. The result shows that all the identified alcohols (except 2-heptanol and 1-octen-3-ol) were affected by the breed, with a significantly higher amount in the Hanwoo meat (*p* < 0.05). Alcohols are known to be the products of the thermal oxidation of fatty acid-derived products during the cooking of meat [50]. Therefore, the higher amounts of the alcohols in Hanwoo meat may be related to its higher IMF content. 1-pentanol, hexanol, 1-hexanol, and 1-heptanol have been found to have desirable odor notes (oily, fatty, and green) in cooked meat [8].

Nitrogen- and sulfur-containing heterocyclic compounds derived from the Mallard reaction between amino acids and reducing sugars, with their low odor detection threshold, have been reported to contribute to the desirable aromas in cooked meat [13]. By using the gas chromatography–olfactometry technique, Frank et al. [12] demonstrated that the pyrazines contribute to the meaty and roasty odor of grilled beef. In the present study, 3 sulfur-containing compounds and 3 pyrazines were detected in the cooked LL muscles from both breeds. These identified compounds have also been found in cooked beef in the literature [15,17]. Out of them, 2 compounds, including methylpyrazine and 2-ethyl-3,5-dimethylpyrazine, were affected by the breed, with the Chikso meat having a higher amount compared to the Hanwoo (*p* < 0.05). The lower amount of identified pyrazines in Hanwoo meat could be due to the effect of fat-derived aldehydes, whose abundance can modulate the Mallard reaction pathway by inhibiting the formation of nitrogen- and sulfur-containing heterocyclic compounds [13,49].

## 4. Conclusions

This study demonstrates that under identical feeding conditions, the breed showed a significant effect on quality attributes such as shear force (tenderness), water holding capacity, color, and taste- and aroma-related components. Some sweetness-related free amino acids (alanine, proline, and threonine) were found at a higher concentration in the Chikso LL muscle, while a higher amount of methionine and glutamine associated with umami taste were found in the Hanwoo SM muscle. A total of 36 metabolites were identified and quantified in the meat samples; out of them, 7 compounds were affected by the breed. Hanwoo meat, with a higher amount of fat-derived aldehydes, may have a higher intensity of fatty aromas, while Chikso meat may be associated with a roasty aroma due to the presence of a higher amount of pyrazines. The insights into the effect of breed on the muscle fiber properties, shelf life, and consumer perception of beef under identical feeding conditions will be investigated in our future study.

## Figures and Tables

**Figure 1 foods-12-00805-f001:**
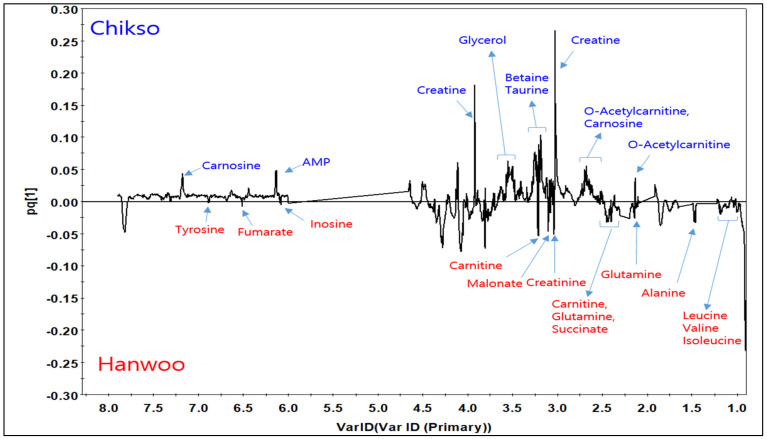
Representative ^1^H NMR spectrum of beef LL muscles from Chikso and Hanwoo. Peaks arising from some common metabolites in the spectrum.

**Table 1 foods-12-00805-t001:** Technological quality traits of LL and SM muscles from Chikso and Hanwoo under identical rearing conditions.

Items	*Longissimus Lumborum* (LL)		*Semimembranosus* (SM)	
	Chikso	Hanwoo	Chikso	Hanwoo
pH	5.74 ± 0.02	5.71 ± 0.04	5.78 ± 0.03	5.70 ± 0.03
Water holding capacity (%)	49.15 ± 2.58 ^a^	42.75 ± 0.59 ^b^	49.49 ± 2.65	44.30 ± 1.46
Cooking loss (%)	18.25 ± 0.43 ^b^	22.73 ± 1.34 ^a^	23.93 ± 0.55 ^b^	26.49 ± 0.92 ^a^
Shear force (kgf)	6.45 ± 0.68 ^a^	3.82 ± 0.27 ^b^	5.82 ± 0.35 ^a^	4.33 ± 0.31 ^b^

Means within a row (in each muscle type) with different superscripts (^a^,^b^) differ significantly (*p* < 0.05).

**Table 2 foods-12-00805-t002:** Color traits of LL and SM muscles from Chikso and Hanwoo under identical rearing conditions.

Items	*Longissimus Lumborum* (LL)		*Semimembranosus* (SM)	
	Chikso	Hanwoo	Chikso	Hanwoo
Lightness (L*)	34.38 ± 0.47 ^b^	38.74 ± 1.28 ^a^	33.91 ± 0.81 ^b^	37.82 ± 1.51 ^a^
Redness (a*)	18.98 ± 0.58 ^b^	23.53 ± 0.64 ^a^	20.94 ± 1.07 ^b^	25.06 ± 0.66 ^a^
Yellowness (b*)	7.72 ± 0.42 ^b^	11.14 ± 0.47 ^a^	8.67 ± 0.72 ^b^	11.75 ± 0.54 ^a^

Means within a row (in each muscle type) with different superscripts (^a^,^b^) differ significantly (*p* < 0.05).

**Table 3 foods-12-00805-t003:** Concentration (mg/100 g) of free amino acid in the LL and SM muscles from Chikso and Hanwoo under identical rearing conditions.

Free Amino Acid	*Longissimus Lumborum* (LL)		*Semimembranosus* (SM)	
	Chikso	Hanwoo	Chikso	Hanwoo
Glycine	5.41 ± 1.41	4.96 ± 2.14	6.91 ± 0.75	6.27 ± 1.64
Alanine	24.06 ± 1.94 ^a^	18.89 ± 4.07 ^b^	24.55 ± 7.09	24.06 ± 4.67
Serine	3.47 ± 0.50	3.15 ± 0.57	3.63 ± 0.58	3.73 ± 0.53
Proline	3.34 ± 0.40 ^a^	2.25 ± 0.51 ^b^	3.45 ± 0.86	3.17 ± 0.69
Valine	4.57 ± 0.82	3.93 ± 0.93	5.57 ± 1.57	5.21 ± 0.52
Threonine	3.87 ± 0.53 ^a^	2.92 ± 0.53 ^b^	4.17 ± 0.92	4.05 ± 0.44
Leucine	8.27 ± 1.33	7.74 ± 2.17	10.17 ± 3.02	10.82 ± 0.81
Isoleucine	2.36 ± 0.77	2.03 ± 0.50	3.08 ± 1.34	3.08 ± 0.33
Aspartate	1.40 ± 0.13	1.33 ± 0.00	1.33 ± 0.00	1.33 ± 0.00
Lysine	4.17 ± 0.87	3.51 ± 0.58	5.41 ± 1.36	4.53 ± 0.70
Glutamic acid	6.63 ± 0.74	6.78 ± 0.83	6.63 ± 1.71	6.70 ± 0.94
Methionine	2.54 ± 0.17	2.61 ± 0.66	2.98 ± 0.24 ^b^	3.51 ± 0.15 ^a^
Histidine	2.48 ± 0.25	2.17 ± 0.25	2.72 ± 0.39	2.48 ± 0.44
Phenylalanine	4.21 ± 0.56	3.88 ± 0.87	4.96 ± 1.18	5.62 ± 0.81
Arginine	6.53 ± 2.39	5.84 ± 1.40	7.84 ± 2.10	6.97 ± 1.17
Tyrosine	4.53 ± 0.47	4.44 ± 0.86	5.35 ± 0.80	5.62 ± 0.36
Cysteine	0.24 ± 0.00	0.24 ± 0.00	0.24 ± 0.00	0.24 ± 0.00
Asparagine	0.66 ± 0.15	0.53 ± 0.22	0.79 ± 0.22	0.86 ± 0.25
Glutamine	20.46 ± 4.87	18.86 ± 7.95	15.64 ± 5.43 ^b^	22.95 ± 12.08 ^a^
Tryptophan	0.92 ± 0.20	0.71 ± 0.20	1.02 ± 0.41	1.12 ± 0.20

Means within a row (in each muscle type) with different superscripts (^a^,^b^) differ significantly (*p* < 0.05).

**Table 4 foods-12-00805-t004:** Concentration (mM/kg) of metabolites in the cooked LL and SM muscles from Chikso and Hanwoo under identical rearing conditions.

Metabolites	*Longissimus Lumborum* (LL)		*Semimembranosus* (SM)	
	Chikso	Hanwoo	Chikso	Hanwoo
3-Hydroxybutyric acid	28.19 ± 6.47	21.77 ± 4.50	35.26 ± 5.82	27.65 ± 6.29
AMP	9.80 ± 4.41	5.60 ± 1.07	9.13 ± 11.18	6.17 ± 2.74
Acetate	2.27 ± 1.30	2.09 ± 0.34	5.15 ± 3.22	2.49 ± 0.31
Alanine	8.05 ± 2.14	11.39 ± 0.73	9.63 ± 2.17	9.73 ± 0.63
Anserine	1.07 ± 0.65	2.32 ± 1.60	2.52 ± 1.53 ^a^	1.16 ± 1.56 ^b^
Betaine	4.50 ± 1.79	2.86 ± 0.44	5.53 ± 0.81	3.19 ± 0.16
Carnitine	9.59 ± 4.79	12.72 ± 1.40	13.30 ± 1.82	11.30 ± 0.87
Carnosine	41.27 ± 10.66	43.27 ± 2.27	58.00 ± 3.66	39.70 ± 3.92
Choline	2.01 ± 0.68	1.57 ± 0.38	3.67 ± 18.58	1.85 ± 4.90
Creatine	118.41 ± 36.80	94.32 ± 5.52	146.43 ± 1.33	108.91 ± 0.54
Creatine phosphate	2.33 ± 1.57	2.37 ± 1.98	3.14 ± 35.12	3.41 ± 25.93
Creatinine	4.00 ± 1.35	7.08 ± 2.38	5.05 ± 3.12	3.82 ± 1.97
Ethanolamine	2.42 ± 0.89	3.38 ± 0.97	3.85 ± 2.17	3.11 ± 0.81
Fumarate	1.03 ± 0.21	1.07 ± 0.74	2.06 ± 1.25	2.01 ± 1.30
Glucose	22.93 ± 10.34	22.09 ± 1.53	30.93 ± 0.57	28.24 ± 0.41
Glutamine	10.77 ± 5.74	11.02 ± 2.79	7.02 ± 8.77	8.79 ± 6.15
Glutathione	1.77 ± 0.85	2.27 ± 0.33	1.81 ± 1.84	1.60 ± 2.46
Glycerol	17.77 ± 3.78	15.09 ± 1.60	28.03 ± 0.44	16.30 ± 0.27
Glycine	5.45 ± 1.74	5.45 ± 1.51	7.60 ± 9.54	7.09 ± 3.23
Hypoxanthine	1.94 ± 0.43 ^b^	5.13 ± 0.70 ^a^	3.40 ± 2.77	6.23 ± 1.24
Inosine	1.75 ± 0.44 ^b^	3.24 ± 0.22 ^a^	2.41 ± 1.82	3.04 ± 1.99
Isoleucine	0.84 ± 0.15	1.12 ± 0.19	1.26 ± 1.25 ^b^	1.88 ± 0.50 ^a^
Lactate	290.00 ± 33.99	261.15 ± 20.91	364.19 ± 0.09	300.43 ± 0.27
Leucine	1.89 ± 0.27	2.76 ± 1.60	1.99 ± 119.88	2.84 ± 51.50
Malonate	8.91 ± 6.30	9.52 ± 0.34	10.05 ± 0.54	10.39 ± 0.44
N,N-Dimethylglycine	0.37 ± 0.16	0.31 ± 0.01	0.45 ± 1.46	0.30 ± 1.94
O-Acetylcarnitine	5.80 ± 0.68 ^a^	4.04 ± 0.26 ^b^	7.73 ± 0.11	4.95 ± 0.05
O-Phosphocholine	2.24 ± 0.88	2.35 ± 0.45	2.91 ± 2.72	2.54 ± 0.98
Phenylalanine	0.74 ± 0.08 ^b^	0.91 ± 0.0 ^a^	0.79 ± 0.64	1.06 ± 0.43
Succinate	2.92 ± 1.11	4.13 ± 0.37	1.67 ± 0.36	1.96 ± 0.18
Taurine	12.24 ± 4.09	8.53 ± 1.40	15.10 ± 1.19	8.58 ± 1.11
Trimethylamine	0.18 ± 0.12	0.13 ± 0.02	0.19 ± 6.55 ^a^	0.09 ± 0.42 ^b^
Tyrosine	0.74 ± 0.21	1.06 ± 0.21	1.04 ± 0.02	1.10 ± 0.02
Valine	1.30 ± 0.50	1.71 ± 0.33	1.83 ± 0.45	2.14 ± 0.12
myo-Inositol	3.13 ± 0.93	3.87 ± 0.81	4.41 ± 0.41	3.28 ± 0.45
sn-Glycero-3-phosphocholine	6.33 ± 4.04	4.35 ± 1.46	5.82 ± 0.13	6.29 ± 1.18

Means within a row (in each muscle type) with different superscripts (^a^,^b^) differ significantly (*p* < 0.05).

**Table 5 foods-12-00805-t005:** Concentration (μg/g) of volatile flavor compounds in cooked *Longissimus lumborum* muscles from Chikso and Hanwoo under identical rearing conditions.

Volatile Flavor Compounds	RT (min)	Chikso	Hanwoo	ID ^(1)^
Aldehydes				
2-methyl pentanal	1.66	0.03 ± 0.03	0.05 ± 0.03	MS + STD
2-methyl propanal	1.827	0.02 ± 0.02	0.01 ± 0.01	MS + STD
3-methyl butanal	2.504	0.03 ± 0.03	0.06 ± 0.04	MS + STD
2-methyl butanal	2.753	0.03 ± 0.03	0.06 ± 0.04	MS + STD
Pentanal	3.041	0.22 ± 0.17 ^b^	0.67 ± 0.12 ^a^	MS + STD
Hexanal	5.65	2.59 ± 1.63 ^b^	5.66 ± 1.13 ^a^	MS + STD
E,2-hexenal	7.37	0.001 ± 0.00 ^b^	0.01 ± 0.00 ^a^	MS + STD
Heptanal	8.799	0.02 ± 0.02 ^b^	0.89 ± 0.29 ^a^	MS + STD
2-Heptenal	10.295	0.00 ± 0.00	0.07 ± 0.14	MS + STD
Benzaldehyde	10.355	0.001 ± 0.00 ^b^	0.03 ± 0.01 ^a^	MS + STD
E,E,2,4-Noadienal	11.183	0.01 ± 0.01 ^b^	0.05 ± 0.02 ^a^	MS + STD
Octanal	11.448	0.33 ± 0.13	0.46 ± 0.14	MS + STD
Benzeneacetaldehyde	12.394	0.001 ± 0.00	0.001 ± 0.00	MS + STD
E,2-Octenal	12.714	0.001 ± 0.00	0.02 ± 0.01	MS + STD
Nonanal	13.713	0.01 ± 0.00 ^b^	0.25 ± 0.12 ^a^	MS + STD
E,2-nonenal	14.829	0.02 ± 0.03	0.03 ± 0.02	MS + STD
Decanal	15.714	0.001 ± 0.00 ^b^	0.01 ± 0.00 ^a^	MS + STD
E,2-Decenal	16.745	0.001 ± 0.00	0.03 ± 0.02	MS + STD
2,4-Dodecadienal	17.324	0.00 ± 0.00	0.00 ± 0.00	MS + STD
Undecenal	17.552	0.001 ± 0.00	0.001 ± 0.00	MS + STD
2-undecenal	18.522	0.00 ± 0.00	0.01 ± 0.01	MS + STD
Alcohols				
2-Heptanol	3.615	0.01 ± 0.01	0.00 ± 0.00	MS + STD
1-Pentanol	4.605	0.12 ± 0.06 ^b^	0.37 ± 0.15 ^a^	MS + STD
Hexanol	5.21	0.001 ± 0.00 ^b^	0.01 ± 0.00 ^a^	MS + STD
1-Hexanol	7.892	0.01 ± 0.01 ^b^	0.10 ± 0.03 ^a^	MS + STD
1-Heptanol	10.665	0.001 ± 0.00 ^b^	0.09 ± 0.04 ^a^	MS + STD
1-Octen-3-ol	10.889	0.01 ± 0.01	0.03 ± 0.01	MS + STD
1-Octanol	12.995	0.001 ± 0.00 ^b^	0.05 ± 0.03 ^a^	MS + STD
Ketones				
2,3-Butanedione	1.988	0.04 ± 0.06	0.06 ± 0.02	MS + STD
2-Butanone	2.026	0.08 ± 0.08	0.09 ± 0.06	MS + STD
2-Heptanone	8.401	0.01 ± 0.01	0.02 ± 0.02	MS + STD
2(5H)-Furanone	9.176	0.05 ± 0.07 ^a^	0.02 ± 0.06 ^b^	MS
Hydrocarbons				
Toluene	4.546	0.00 ± 0.00	0.00 ± 0.01	MS + STD
Ethylbenzene	7.563	0.001 ± 0.00 ^b^	0.02 ± 0.02 ^a^	MS + STD
1,3-dimethylbenzene	7.81	0.01 ± 0.00 ^b^	0.03 ± 0.02 ^a^	MS + STD
5-ethyl-2-methyloctane	12.274	0.00 ± 0.00	0.00 ± 0.00	MS
2,6-dimethyloctane	12.612	0.01 ± 0.00	0.02 ± 0.01	MS
2,5,9-trimethyldecane	12.831	0.00 ± 0.00	0.01 ± 0.00	MS
4,8-dimethyldecane	13.062	0.00 ± 0.00	0.01 ± 0.01	MS
Dodecane	15.591	0.00 ± 0.00	0.01 ± 0.01	MS + STD
Tetredecane	19.088	0.00 ± 0.00	0.00 ± 0.00	MS + STD
Sulfur-and nitrogen-containing compounds				
Methanethiol	1.489	0.01 ± 0.01	0.01 ± 0.01	MS + STD
Carbon sulfide	1.688	0.01 ± 0.02	0.01 ± 0.00	MS + STD
Carbon disulfide	1.761	0.01 ± 0.02	0.01 ± 0.01	MS + STD
Methylpyrazine	6.554	0.04 ± 0.09 ^a^	0.001 ± 0.00 ^b^	MS + STD
2,5-dimethylpyrazine	9.024	0.01 ± 0.02	0.01 ± 0.00	MS + STD
2-Ethyl-3,5-dimethylpyrazine	13.194	0.01 ± 0.01 ^a^	0.001 ± 0.01 ^b^	MS + STD

Means within the same row with different superscripts (^a^, ^b^) are significantly different (*p* < 0.05). ID ^(1)^: Identification method: The compounds were identified by mass spectra (MS) from a library or external standards (STD).

## Data Availability

Data is contained within the article.

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
