# Peer review of "A Comparative Study on the Meat Quality, Taste and Aroma Related Compounds between Korean Hanwoo and Chikso Cattle"

_foods, 2023, doi:10.3390/foods12040805_

Round 1

Reviewer 1 Report

The authors have submitted an article that outlines an investigation on the differences of Korean Hanwoo and Chikso cattle breeds in meat quality, taste-and and aroma flavor-related compounds.

The title and keywords accurately reflect the content of the manuscript. The authors of this manuscript gave us a clear introduction to the study they have done with an overall good literature review. Research aim of the study is clearly defined. The materials and methods are described clearly with sufficient details of the performed measurements and the measurement techniques are appropriate to resolve the stated objectives of the study. However, there are not enough samples per group. Therefore, I think that additional analyses should be performed to improve sample size and to obtain more reliable results. Despite the fact that my mother language in not English, manuscript should be checked by native speaker. After improving the manuscript in these aspects, it can be submitted for consideration in foods journal.

According to my opinion, the manuscript should not be accepted in the present form for publication in foods journal.

Author Response

The comments raised by the reviewer is replied as shown in the PDF file (please check the attached file)

Reviewer 2 Report

Article foods- 2189299: ‘A comparative study on the meat quality, taste-and and aroma flavor-related compounds between Korean Hanwoo and Chikso cattle’

The authors propose to study the differences in meat quality, and taste- and aroma flavor-related components of Chikso beef, versus those of Hanwoo beef under identical raising conditions.

I found the objectives of the work interesting and worthy of being researched. In my opinion, the research was well conducted, the material and methods are adequate to achieve the objectives proposed but need some improvement in the description of the methodology used, the results are presented appropriately and the discussion is well conducted with current cited references, but needs some clarification.

Some comments:

Title - I have some difficulty understanding the title ‘‘taste-and and aroma flavor-related compounds’ Since Flavor is the blend of taste and smell sensations evoked by a substance in the mouth, One possibility is ‘taste and aroma flavor related compounds.

Line 13 - ‘longissimus lumborum(LL) and semimembranosus(SM); add a space after ‘lumborum’ and ‘semimembranosus’.

Line 45 – ‘(free amino acids and metabolites etc.)’; I would recommend not using etc. in an academic paper, see other cases in the text.

Line 55 – ‘Longissimus lumborum (LL) and Semimembranosus (SM)’; uniformize the writing form with abstract and text (in text LL and SM are not italicized).

Line 69 – ‘D65 illuminant*C’; I don’t understand which illuminant was used: D65 or C? Meat trays must be covered with PE film to avoid meat drying and 60 min blooming is also advisable.

Line 71 – ‘CIE L* (lightness), CIE a* (redness), and CIE b* (yellowness).’; the use of CIE and the asterisk is, in my opinion, redundant because the asterisk means that the space used is the one from CIE, so use CIE L or L*; the same for the other two coordinates.

Line 72 – I think that the pH was measured directly in meat, if so, please add this information.

Line 77 – ‘in a preheated water bath until the core temperature reached 72 °C.’; as the rate of meat cooking may affect WHC please indicate the water bath temperature.

Line 82 – ‘WBSF values (kg-force) were obtained’; kgf does not comply with the International System of Units (SI), please convert the values to Newton (N).

Line 93 – ‘The content of FFAs’, I think that is FAAs, please in the title of sub-chapter ‘2.5. Tastes-related compounds (free amino acids and metabolites)’ add the acronym FAAs.

Line 95 – ‘1200 × g for’; g in italic.

Line 102 – ‘The UPLC conditions used were the same as those described by Cho et al. [16].’; please, briefly describe the UPLC conditions.

Line 104 – ‘(2-cm thick steak form)’, remove the hyphen.

Line 126 – indicate how the results were expressed.

Regarding the Statistical analysis, I have some questions:

1. The authors refer that the breed was considered as a fixed factor, omitting relative to the muscle, which I believe to be a random effect.

2. Why was it necessary to use the test for multiple comparisons of means (Duncan's Multiple Range Test), if only two means are compared? Doesn't the analysis of the variance chart already give the level of significance for the breed effect and the muscle effect?

Line 131 – ‘Data were presented as means ± standard.’; the authors mean standard deviation? Please rectify.

Line 136 - ‘postmortem’ but in material and methods the authors use post-mortem, please uniformize in all the article.

Line 140-141 – ‘It was reported that any abnormal rate of postmortem glycolysis may result in inferior meat quality such as; a low (<5.5) and high (>5.8) ultimate pH is associated with pale and dark-firm and dry (DFD) beef, respectively [19,20].’; I have some questions about this sentence.

There is no doubt that a high final pH results in a quality defect called DFD that can occur in both beef and pork. Regarding the relationship that the authors refer to between pH<5.5 and the meat paleness, it is not evident to me, because: (1) if the authors are referring to an ultimate pH lower than 5.5 and its relationship with the pale color, as far as I know, it occurs in the Hampshire pig breed mainly in the females because they have high reserves of glycogen. (2) if they are referring to the postmortem pH decrease rate, this is the case for PSE meat in pigs. In these meats at 45 min - 1 h postmortem the pH can present values lower than 5.8 and possibly even near the ultimate pH of the meat. In cattle, the PSE condition is not expressed. (3) In the article the authors refer to this article ‘Viljoen, H.F.; de Kock, H.L.; Web, E.C. 2002. Consumer acceptability of dark, firm, and dry (DFD) and normal pH beef steaks. 332 Meat Sci. 2002, 61, 181–185. 333‘ but I do not see in the article any relationship described between pH <5.5 and the paleness of the meat. Considering these questions, I would appreciate the clarification of the authors and the necessary corrections in the text, if applicable.

Line 161-164 – ‘The lower shear force values in the Hanwoo beef may be attributed to its lower intramuscular fat (IMF) content compared to the Chikso (our analysis showed that the IMF was approximately 3 times higher in Hanwoo compared to the Chikso, data not shown) because the level of IMF is negatively correlated to the WBSF values [26,27].’ I think that the authors mean The lower shear force values in the Hanwoo beef may be attributed to its HIGHER intramuscular fat (IMF) content compared to the Chikso’.

I agree with this conclusion but there is the same relationship between cooking losses and WBSF, and Hanwoo have higher cooking losses. I think that the authors should mention this inconsistency.

Concerning meat color, I think that the higher L* and b* and lower a* values of Hanwoo may be mainly related to the higher IMF referred by the authors.

Table 3 – add ‘Free amino acids’ as the head in the first column.

Table 4 – add ‘Metabolites’ as the head in the first column.

Table 5 – replace ‘items’ with ‘Volatile flavor compounds’

Author Response

Response to reviewer’s comment

Dear the valuable reviewers!

Thank you so much for your great efforts in reviewing our manuscript. Based on your comments and recommendations, the authors have drastically revised the manuscript. The places where the authors revised/changed are marked with green color.

Article foods- 2189299: ‘A comparative study on the meat quality, taste-and and aroma flavor-related compounds between Korean Hanwoo and Chikso cattle’

The authors propose to study the differences in meat quality, and taste- and aroma flavor-related components of Chikso beef, versus those of Hanwoo beef under identical raising conditions.

I found the objectives of the work interesting and worthy of being researched. In my opinion, the research was well conducted, the material and methods are adequate to achieve the objectives proposed but need some improvement in the description of the methodology used, the results are presented appropriately and the discussion is well conducted with current cited references, but needs some clarification.

Re: Thank you very much for your thoughtful comments

Some comments:

Title - I have some difficulty understanding the title ‘‘taste-and and aroma flavor-related compounds’ Since Flavor is the blend of taste and smell sensations evoked by a substance in the mouth, One possibility is ‘taste and aroma flavor related compounds.

Re: Thank you very much for pointing out. The title was edited according to the reviewer’s suggestion. Please check line 1-2 (marked with green color) in the revised manuscript

 Line 13 - ‘longissimus lumborum(LL) and semimembranosus(SM); add a space after ‘lumborum’ and ‘semimembranosus’.

Re: Thank you very much for pointing out. A space after ‘lumborum’ and ‘semimembranosus’ was added. Please check line 14 (marked with green color) in the revised manuscript

Line 45 – ‘(free amino acids and metabolites etc.)’; I would recommend not using etc. in an academic paper, see other cases in the text.

Re: Thank you very much for pointing out. The term “etc.” was removed throughout the revised manuscript. Please check line 47 (marked with green color) in the revised manuscript

Line 55 – ‘Longissimus lumborum (LL) and Semimembranosus (SM)’; uniformize the writing form with abstract and text (in text LL and SM are not italicized).

Re: Thank you very much for pointing out. ‘Longissimus lumborum (LL) and Semimembranosus (SM) were uniformized throughout the revised manuscript. Please check line 64 (marked with green color) in the revised manuscript

Line 69 – ‘D65 illuminant*C’; I don’t understand which illuminant was used: D65 or C? Meat trays must be covered with PE film to avoid meat drying and 60 min blooming is also advisable.

Re: Thank you very much for pointing out. It was mistyped. The correct one is D65 illuminant. For many years, our research group usually bloom the fresh meat for about 30 min after cutting, and during the blooming period the meat is exposed to air environment (without covering) to get oxygen for the cherry-red color development. This is the common practical condition in the meat research. Please check line 71 (marked with green color) in the revised manuscript

Line 71 – ‘CIE L* (lightness), CIE a* (redness), and CIE b* (yellowness).’; the use of CIE and the asterisk is, in my opinion, redundant because the asterisk means that the space used is the one from CIE, so use CIE L or L*; the same for the other two coordinates.

Re: Thank you very much for pointing out. According to the reviewer’s comment, the author has revised the statement. Please check line 73 (marked with green color) in the revised manuscript

Line 72 – I think that the pH was measured directly in meat, if so, please add this information.

Re: Thank you very much for pointing out. The information was included into the revised manuscript. Please check line 74-75 (marked with green color) in the revised manuscript

Line 77 – ‘in a preheated water bath until the core temperature reached 72 °C.’; as the rate of meat cooking may affect WHC please indicate the water bath temperature.

Re: Thank you very much for pointing out. The water bath temperature was mentioned. Please check line 80 (marked with green color) in the revised manuscript

Line 82 – ‘WBSF values (kg-force) were obtained’; kgf does not comply with the International System of Units (SI), please convert the values to Newton (N).

Re: Thank you very much for pointing out. For decades, our meat research group usually use the unit “kg-force” when measuring the WBSF values of meat. And we also used this unit in many published papers in Journals. Many other scientists also use the unit of WBSF value as “kgf”. In this case, it will be hard for the authors to replace by the Newton unit because in the results and discussion section the authors used data from other studies that also used the same WBSF unit (kgf) to compare with our results.

Line 93 – ‘The content of FFAs’, I think that is FAAs, please in the title of sub-chapter ‘2.5. Tastes-related compounds (free amino acids and metabolites)’ add the acronym FAAs.

Re: Thank you very much for pointing out. The information was corrected. Please check line 98-99 (marked with green color) in the revised manuscript

Line 95 – ‘1200 × g for’; g in italic.

Re: Thank you very much for pointing out. “g” was in italic form. Please check line 101 (marked with green color) in the revised manuscript

Line 102 – ‘The UPLC conditions used were the same as those described by Cho et al. [16].’; please, briefly describe the UPLC conditions.

Re: Thank you very much for pointing out. The UPLC condition was briefly described in the revised manuscript. Please check line 108-110 (marked with green color) in the revised manuscript

Line 104 – ‘(2-cm thick steak form)’, remove the hyphen.

Re: Thank you very much for pointing out. The hyphen was removed. Please check line 112 (marked with green color) in the revised manuscript

Line 126 – indicate how the results were expressed.

Re: Thank you very much for pointing out. The expression of results was mentioned. Please check line 136 (marked with green color) in the revised manuscript

Regarding the Statistical analysis, I have some questions:

  1. The authors refer that the breed was considered as a fixed factor, omitting relative to the muscle, which I believe to be a random effect.

Re: Thank you very much for pointing out. In this study, the statistical analysis was done separately for each muscle type. Only one factor was the breed. The reason why the authors did not compare the muscle type between breeds was that these two muscles are quite different from anatomical location, physicochemical composition and quality as well. Thus, it is not necessary to compare the results between two muscle types. Please check line 138-139 (marked with green color) in the revised manuscript

  1. Why was it necessary to use the test for multiple comparisons of means (Duncan's Multiple Range Test), if only two means are compared? Doesn't the analysis of the variance chart already give the level of significance for the breed effect and the muscle effect?

Re: Thank you very much for pointing out. Actually, our research group usually uses this model when comparing means and it still works well and gives a correct result. 

Line 131 – ‘Data were presented as means ± standard.’; the authors mean standard deviation? Please rectify.

Re: Thank you very much for pointing out. It was changed to “standard deviation”. Please check line 142 (marked with green color) in the revised manuscript

Line 136 - ‘postmortem’ but in material and methods the authors use post-mortem, please uniformize in all the article.

Re: Thank you very much for pointing out. It was changed to “post-mortem” throughout the manuscript. Please check line 148 (marked with green color) in the revised manuscript

Line 140-141 – ‘It was reported that any abnormal rate of postmortem glycolysis may result in inferior meat quality such as; a low (<5.5) and high (>5.8) ultimate pH is associated with pale and dark-firm and dry (DFD) beef, respectively [19,20].’; I have some questions about this sentence.

There is no doubt that a high final pH results in a quality defect called DFD that can occur in both beef and pork. Regarding the relationship that the authors refer to between pH<5.5 and the meat paleness, it is not evident to me, because: (1) if the authors are referring to an ultimate pH lower than 5.5 and its relationship with the pale color, as far as I know, it occurs in the Hampshire pig breed mainly in the females because they have high reserves of glycogen. (2) if they are referring to the postmortem pH decrease rate, this is the case for PSE meat in pigs. In these meats at 45 min - 1 h postmortem the pH can present values lower than 5.8 and possibly even near the ultimate pH of the meat. In cattle, the PSE condition is not expressed. (3) In the article the authors refer to this article ‘Viljoen, H.F.; de Kock, H.L.; Web, E.C. 2002. Consumer acceptability of dark, firm, and dry (DFD) and normal pH beef steaks. 332 Meat Sci. 2002, 61, 181–185. 333‘ but I do not see in the article any relationship described between pH <5.5 and the paleness of the meat. Considering these questions, I would appreciate the clarification of the authors and the necessary corrections in the text, if applicable.

Re: Thank you for your thoughtful comment. I think that we all well know about the normal and abnormal pH of meat and how it affects meat quality. The information (cited from previous studies) that the authors mentioned in the text simply is an example that the authors would like to say the factors affecting ultimate pH of meat decline and its effects on meat quality. In our study, no differences in pH was found between the breeds. Regarding the reviewer’s comment, the statement was revised to make thing more lucid in the text. Please check line 149-150 (marked with green color) in the revised manuscript

Line 161-164 – ‘The lower shear force values in the Hanwoo beef may be attributed to its lower intramuscular fat (IMF) content compared to the Chikso (our analysis showed that the IMF was approximately 3 times higher in Hanwoo compared to the Chikso, data not shown) because the level of IMF is negatively correlated to the WBSF values [26,27].’ I think that the authors mean ‘The lower shear force values in the Hanwoo beef may be attributed to its HIGHER intramuscular fat (IMF) content compared to the Chikso’.

I agree with this conclusion but there is the same relationship between cooking losses and WBSF, and Hanwoo have higher cooking losses. I think that the authors should mention this inconsistency.

Re: Thank you for pointing out. We missed in this information. The statement was corrected. Please check line 169 (marked with green color) in the revised manuscript

Concerning meat color, I think that the higher L* and b* and lower a* values of Hanwoo may be mainly related to the higher IMF referred by the authors.

Re: Thank you, this information was also mentioned in the text. Please check line 184-186 (marked with green color) in the revised manuscript

Table 3 – add ‘Free amino acids’ as the head in the first column.

Re: Thank you for pointing out. ‘Free amino acids’ was added. Please check line 211 (marked with green color) in the revised manuscript

Table 4 – add ‘Metabolites’ as the head in the first column.

Re: Thank you for pointing out. ‘Metabolites’ was added. Please check line 235(marked with green color) in the revised manuscript

Table 5 – replace ‘items’ with ‘Volatile flavor compounds’

Re: Thank you for pointing out. ‘Volatile flavor compounds’ was replaced. Please check line 263 (marked with green color) in the revised manuscript

Once again, thank you very much for your efforts in improving the quality of our manuscript.

Reviewer 3 Report

The investigators compared meat quality, free amino acids, metabolites and volatile compounds between Chilkso and Hanwoo beef. The investigators performed a lot of examination and layout a nice manuscript. However, in my opinion, there are some points that must be clarified/modified. Also, some typos and minor grammatical errors are found. 

1) Abstract:

- line 12: steers "that" were... 

- Please state sample size.

2) Introduction & objective

- The investigators already provided an introduction about Chikso and indicated that the study about Chikso meat quality is still limited. However, the rationale of this study is not clear to me. The manuscript will be remarkably strengthened if the investigators state the significant impact of this study.

2) M&M:

- line 62: The number of sample size is not clear to me.  Not sure if n = 7 means there is 7 pieces of meat per breeds (and only 1 animal per breed was used). Or, the samples were collected from 7 different animals per breed. This must be clarified.

- line 81: how did you prepare the strips? Cork them out?

- line 87: Please state how the investigators prepared the samples; cut into cubes or else? Also, I assume that the investigators monitored that the meat was under the water during the entire cooking and cooling?

- line 90-91: Please consider adding mathematical formula for calculating WHC. This may help the readers to understand how WHC was calculated and how MC was associated with WHC as now it's difficult to follow the text.

-line 104: Please discuss on why the different thickness of steak between cook loss and metabolite analyses was used.

-line 117: Please discuss on why chopped steak was used. Why wouldn't the steak be cooked in the same manner for determination of cook loss, metabolites and volatile compounds. If using chopped samples, would the volatile compounds be lost from the samples during cooking?

- line 114: In my opinion, the term "volatile compounds" would be more appropriate than "aroma volatile compounds" as not all volatiles related to aroma. Also, no correlation between the identified compounds and aroma was directly done in this study.

line 128: Please indicate how the data from different muscle type was analyzed (separately?)

4) Results and discussion

- line 162: "lower intramuscular fat" -- should it be "higher"?

- line 176-177; I strongly recommend the investigators to include those data, particularly chemical composition and myoglobin content, in the manuscript as they will be helpful for the discussion and will significantly strengthened this manuscript.

Author Response

Response to reviewer’s comment

Dear the valuable reviewers!

Thank you so much for your great efforts in reviewing our manuscript. Based on your comments and recommendations, the authors have drastically revised the manuscript. The places where the authors revised/changed are marked with yellow color.

The investigators compared meat quality, free amino acids, metabolites and volatile compounds between Chilkso and Hanwoo beef. The investigators performed a lot of examination and layout a nice manuscript. However, in my opinion, there are some points that must be clarified/modified. Also, some typos and minor grammatical errors are found. 

1) Abstract:

- line 12: steers "that" were... 

Re: Thank you for pointing out. The authors assume that the sentence is in correct state.

- Please state sample size.

Re: Thank you for pointing out. The sample size was mentioned in the text.  Please check line 13 (marked with yellow color) in the revised manuscript.

2) Introduction & objective

- The investigators already provided an introduction about Chikso and indicated that the study about Chikso meat quality is still limited. However, the rationale of this study is not clear to me. The manuscript will be remarkably strengthened if the investigators state the significant impact of this study.

Re: Thank you for your thoughtful comment. Indeed, study on the Chikso’s meat quality is too limited and how the differences in its meat quality with those from other commercial cattle breeds have not been compared. Basically, the authors may focus on writing the effects of breed on the meat quality in the Introduction section. However, the topic of effect of breed on meat quality is too old not attractive to readers any more. Therefore, main objective of this study was to compare the quality characteristics between the Chikso cattle breed and Hanwoo breeds as mentioned in the Introduction section. Please check line 51-55 (marked with yellow color) in the revised manuscript

2) M&M:

- line 62: The number of sample size is not clear to me.  Not sure if n = 7 means there is 7 pieces of meat per breeds (and only 1 animal per breed was used). Or, the samples were collected from 7 different animals per breed. This must be clarified.

Re: Thank you for pointing out. In this study, we used 14 animals (the body weight of around 680 – 740 kg per animal; 7 animals per breed). The animals were reared under identical condition and collected at 30 months of age. After slaughter at the abattoir in my institute, whole longissimus lumborum (LL, n=7) and semimembranosus (SM, n=7) muscles from each cattle breed were collected. Please check line 57, 64-65 (marked with yellow color) in the revised manuscript.

- line 81: how did you prepare the strips? Cork them out?

Re: Thank you for pointing out. The strips were made using a 0.5-inch metal corer. This information was mentioned in the text. Please check line 85 (marked with yellow color) in the revised manuscript.

- line 87: Please state how the investigators prepared the samples; cut into cubes or else? Also, I assume that the investigators monitored that the meat was under the water during the entire cooking and cooling?

Re: Thank you for pointing out. The sample preparation for WHC was mentioned in the text. Please check line 88-89 (marked with yellow color) in the revised manuscript.

- line 90-91: Please consider adding mathematical formula for calculating WHC. This may help the readers to understand how WHC was calculated and how MC was associated with WHC as now it's difficult to follow the text.

Re: Thank you for pointing out. The formulation for WHC calculation was added. Please check line 94-96 (marked with yellow color) in the revised manuscript.

-line 104: Please discuss on why the different thickness of steak between cook loss and metabolite analyses was used.

Re: Thank you for pointing out. That was mistyped. The information was corrected. Please check line 112 (marked with yellow color) in the revised manuscript.

-line 117: Please discuss on why chopped steak was used. Why wouldn't the steak be cooked in the same manner for determination of cook loss, metabolites and volatile compounds. If using chopped samples, would the volatile compounds be lost from the samples during cooking?

Re: Thank you for pointing out. The reason why the authors used the chopped samples is that we wish to facilitate/strengthen the chemical reactions (e.g., Mallard reaction) of volatiles formation during cooking because the ground samples may have more chance to contact the frying fan surface and temperature (the reactions for volatiles formation largely rely on the cooking temperature). The information was mentioned in the text. Please check line 124-126 (marked with yellow color) in the revised manuscript.

- line 114: In my opinion, the term "volatile compounds" would be more appropriate than "aroma volatile compounds" as not all volatiles related to aroma. Also, no correlation between the identified compounds and aroma was directly done in this study.

Re: Thank you for pointing out. In combination with the other reviewer’s comment the authors decide to use the term of "volatile flavor compounds" throughout the manuscript. Please check line 122, 238 (marked with yellow color) in the revised manuscript.  

line 128: Please indicate how the data from different muscle type was analyzed (separately?)

Re: Thank you for pointing out. Yes, we analyzed separately for each muscle because only one factor “breed or cattle type” was used. Please check line 138-139 (marked with yellow color) in the revised manuscript.

4) Results and discussion

- line 162: "lower intramuscular fat" -- should it be "higher"?

Re: Thank you for pointing out. We mistyped. It was changed to “higher”. Please check line 169 (marked with yellow color) in the revised manuscript.

- line 176-177; I strongly recommend the investigators to include those data, particularly chemical composition and myoglobin content, in the manuscript as they will be helpful for the discussion and will significantly strengthened this manuscript.

Re: Thank you for your thoughtful comment. Unfortunately, in this research project the experiments on the myoglobin content were not included, and we are sorry to say that now we do not have samples for these experiments. For the chemical composition, its results have already been included in another manuscript that is under considered for publication in a MDPI journal so we cannot add into this paper.

Once again, thank you very much for your efforts in improving the quality of our manuscript.

Reviewer 4 Report

The current study reports an interesting topic that points out the “A comparative study on the meat quality, taste-and and aroma flavor-related compounds between Korean Hanwoo and Chikso cattle”. The manuscript's presentation is adequate. Also, all tables mentioned within the manuscript are provided and fit the presented findings and subject. This is an interesting work. However, minor revised need to be improved in this manuscript before its publication.

Materials and Methods:

In "material and methods" part, 2.1, 2.2, 2.3, 2.4, 2.5 , 2.6, 2.7,  Please be consistent with 2.5 format. Please check the whole article!

Results

Line 147: Is the font size consistent with the whole article?

References:

I suggest avoiding lumping the references. Each reference should be discussed separately or deleted older.

Please, reduce the reference list. Consider staying newer ones.

Author Response

Response to reviewer comment

Dear the valuable reviewers!

Thank you so much for your great efforts in reviewing our manuscript. Based on your comments and recommendations, the authors have drastically revised the manuscript. The places where the authors revised/changed are marked with purple color.

The current study reports an interesting topic that points out the “A comparative study on the meat quality, taste-and and aroma flavor-related compounds between Korean Hanwoo and Chikso cattle”. The manuscript's presentation is adequate. Also, all tables mentioned within the manuscript are provided and fit the presented findings and subject. This is an interesting work. However, minor revised need to be improved in this manuscript before its publication.

Re: Thank you for your comment.

Materials and Methods:

 In "material and methods" part, 2.1, 2.2, 2.3, 2.4, 2.5 , 2.6, 2.7,  Please be consistent with 2.5 format. Please check the whole article!

Re: Thanks for pointing out. The sub-sections from 2.1 to 2.7 was made to a consistent format of 2.5

Results

Line 147: Is the font size consistent with the whole article?

Re: Thanks for pointing out. the font size of corrected. Please check line 156 (marked purple color) in the revised manuscript.

References:

I suggest avoiding lumping the references. Each reference should be discussed separately or deleted older.

Please, reduce the reference list. Consider staying newer ones.

Re: Thanks for pointing out. The authors assume that all the cited references are important in this manuscript. The total number of cited references is similar lesser than in other published papers in Foods. The authors would like to keep these references in the manuscript. 

Once again, thank you very much for your efforts in improving the quality of our manuscript.

Round 2

Reviewer 1 Report

Authors included all suggestions by reviewers, which further improve the quality of manuscript. In addition, they explained the reasons for low sample size with only 14 cattle included in the experiment.

Author Response

Authors included all suggestions by reviewers, which further improve the quality of manuscript. In addition, they explained the reasons for low sample size with only 14 cattle included in the experiment.

Re: Once again, thank you very much for your efforts in improving the quality of our manuscript.

Reviewer 2 Report

The authors adequately answered the questions raised and made the suggested changes.

Author Response

Comments and Suggestions for Authors

The authors adequately answered the questions raised and made the suggested changes.

Re: Once again, thank you very much for your efforts in improving the quality of our manuscript.

Reviewer 3 Report

- Line 57: "A total of 14 steers (7 for Hanwoo, 7 for Chikso)" were used in the present investigation.

- Line 72-74: I am confused whether the investigators used CIE L*, a* b* system or L, a, b. In case of the first, you may want to add * after each parameter.

- WHC: Thank you for stating that the meat was ground. In this case, please add how many technical replicates were used for this experiment. In my experience, it is quite difficult for only 0.5 g of ground samples would well represent the whole beef WHC unless the sample size and/or technical replication are large enough.

- line 124: Thank you for your clarification. Please indicate whether the meat was stirred during the 2-min frying or not. Or, the ground beef were cooked in the form of meat patty? In my opinion, without the stirring, your ultimate aim of using ground beef might not be as effective as you expected. On the other hand, different stirring would let the Maillard reaction developed differently among samples. Also, please state how the steaks were cooked (line 112) -- 4 min for each side or else.

- Volatile flavor compounds: I am still not comfortable about "flavor" as there is no direct experiment in your work to relate the identified compounds and their sensation (using GC-O or trained sensory panel). At the end, it's up to the investigators to confirm and conclude their scientific study and manuscript.

- According to my last comment, thank you for your clarification. I'd suggest to wait for one of your manuscript is accepted; perhaps the one with chemical composition, then you can cite such study in this manuscript. This will be helpful for your readers as well as for yourselves during the discussion.

Author Response

Response to reviewer’s comment

Dear the valuable reviewers!

Thank you so much for your great efforts in reviewing our manuscript. Based on your comments and recommendations, we have drastically revised the manuscript. The places where the authors revised/changed are marked with yellow color.

Comments and Suggestions for Authors

- Line 57: "A total of 14 steers (7 for Hanwoo, 7 for Chikso)" were used in the present investigation.

Re: Thank you for your thoughtful comment. It was changed to "A total of 14 steers (7 for Hanwoo, 7 for Chikso)" were used in the present investigation”. Please check line 57 marked with yellow color in the revised manuscript.

- Line 72-74: I am confused whether the investigators used CIE L*, a* b* system or L, a, b. In case of the first, you may want to add * after each parameter.

Re: Thank you for pointing out. In this case, I also think that you confused. The correct one should be CIE L*, a* b*. The authors already added “ * ” for each the color parameter. Please check line 73 marked with yellow color in the revised manuscript.

- WHC: Thank you for stating that the meat was ground. In this case, please add how many technical replicates were used for this experiment. In my experience, it is quite difficult for only 0.5 g of ground samples would well represent the whole beef WHC unless the sample size and/or technical replication are large enough.

Re: Thank you for your thoughtful comment. I deeply understand your opinion. In our study, each sample was measured in duplicates (2 times replication). This was stated in the revised manuscript.  Please check line 88 marked with yellow color in the revised manuscript.

- line 124: Thank you for your clarification. Please indicate whether the meat was stirred during the 2-min frying or not. Or, the ground beef were cooked in the form of meat patty? In my opinion, without the stirring, your ultimate aim of using ground beef might not be as effective as you expected. On the other hand, different stirring would let the Maillard reaction developed differently among samples. Also, please state how the steaks were cooked (line 112) -- 4 min for each side or else.

Re: Thank you for your thoughtful comment. In our study, the ground beef samples were stirred throughout the 2-min frying. And the steaks were turned at 1-min interval during cooking. This information was stated in the revised manuscript. Please check line 112-113 and 126 marked with yellow color in the revised manuscript.

- Volatile flavor compounds: I am still not comfortable about "flavor" as there is no direct experiment in your work to relate the identified compounds and their sensation (using GC-O or trained sensory panel). At the end, it's up to the investigators to confirm and conclude their scientific study and manuscript.

Re: Thank you for your thoughtful comment. The “Flavor” is the commonly-used term in many studies on volatile compounds in foods (e.g., meat, breads). In our case, we have used this term in tens of studies that have been published in popular journals (Meat Science, LWT-FS & Technology, Food control etc.). The authors assume that there is no problem with the use of “Flavor” term

- According to my last comment, thank you for your clarification. I'd suggest to wait for one of your manuscript is accepted; perhaps the one with chemical composition, then you can cite such study in this manuscript. This will be helpful for your readers as well as for yourselves during the discussion.

Re: Thank you for your thoughtful comment. As we known, in research articles in many cases, authors may mention their analytic results like “data not shown”. It is a very common case. In our case, this study was funded by the Korea government, and the use of data (analytic results) was planned for each stage and articles. Currently, the chemical analysis results were used in another manuscript that we cannot know when it will be accepted. It is hard for the authors to wait until that manuscript is accepted. Please understand for our situation.

Once again, thank you very much for your efforts in improving the quality of our manuscript.
